# Probabilistic Approach to Determine the Spatial Distribution of the Antecedent Moisture Conditions for Different Return Periods in the Atlántico Region, Colombia

Julio Jose Salgado-Cassiani [1], Oscar E. Coronado-Hernández [1,*], Gustavo Gatica [2], Rodrigo Linfati [3] and Jairo R. Coronado-Hernández [4]

1 Facultad de Ingeniería, Universidad Tecnológica de Bolívar, Cartagena de Indias 131001, Colombia; jjsalgado0102@gmail.com
2 Faculty of Engineering, Universidad Andres Bello, Santiago de Chile 7500971, Chile; ggatica@unab.cl
3 Department of Industrial Engineering, Universidad del Bío-Bío, Concepción 4030000, Chile; rlinfati@ubiobio.cl
4 Departamento de Gestión Industrial, Agroindustrial y Operaciones, Universidad de la Costa, Barranquilla 080001, Colombia; jcoronad18@cuc.edu.co
* Correspondence: ocoronado@utb.edu.co; Tel.: +57-301-371-5398

**Abstract:** Previous soil moisture conditions play an important role in the design of hydraulic structures because they are directly related to the runoff threshold associated with a return period. These represent one of the main determinants of the runoff response of a drainage basin. One of the main difficulties facing hydrologists in Colombia lies in the time spent gathering and analyzing information related to the selection of antecedent moisture conditions. In this study, complete records from 19 rainfall stations located in the Atlántico region, Colombia, were used to analyze the cumulative precipitation during the 5 days prior to the annual maximum daily precipitation associated with different return periods using the Gev, Gumbel, Pearson Type III and Log Pearson Type III probability distributions. Different interpolation methods (IDW, kriging and spline) were applied to evaluate the spatial distribution of the antecedent moisture conditions. The main contribution of this research is establishing, using a probabilistic approach, the behavior of antecedent moisture conditions in a particular region, which can be used by engineers and designers to plan water infrastructure. This probabilistic approach was applied to a case study of the Atlántico region, Colombia, where the spatial distribution of antecedent moisture conditions was calculated for several return periods. The results indicate that the better results were obtained with the IDW interpolation method, and the Pearson Type III and Gumbel distributions also showed the best fits based on the Akaike criterion.

**Keywords:** precipitation; frequency analysis; return period; antecedent moisture condition

## 1. Introduction

The hydrological response of a drainage system is subject to several parameters that are related to each other and regulate hydrological processes; for this reason, predicting the hydrological responses of a drainage basin is essential for different purposes, from storm surges to assessing the impacts resulting from land-cover changes altering the water cycle. The spatiotemporal variations in rainfall, the morphometric characteristics of the basins, the physical properties of the soils, the presence and density of the vegetation cover and the antecedent moisture conditions are the most representative factors [1,2].

The antecedent moisture conditions of the soil play an important role in the design of hydraulic structures because they are directly related to the amount of runoff that can be generated for different return periods, which is one of the main conditioning factors for the runoff response of a drainage basin [3,4]. The Natura Resource Conservation Service (NRCS) of the United States (1972) [1] developed the curve number infiltration method,

which allows estimating direct runoff, taking into account the characteristics of the soil, its use and vegetation cover. From the maximum 24 h rainfall records, the type of antecedent moisture can be established. The moisture condition (AMC II) corresponds to the average moisture of the soil. If there are high rainfall intensities during the 5 days prior to the most intense downpour, saturated soil conditions will occur, which is known as antecedent moisture condition type III (AMC III). Otherwise, if rainfall intensities are low, the soil will have the capacity to infiltrate a significant percentage of the direct runoff (AMC I). The AMC plays an important role in the rainfall-triggered shallow landslides [5–7]. Table 1 shows the classification of the antecedent moisture conditions as a function of the total 5-day antecedent rainfall for the type of rainfall station.

**Table 1.** Classification of antecedent moisture classes (AMC) for the soil conservation service (SCS) rainfall abstractions method.

| AMC Group | 5-Day Total Antecedent Rainfall (mm) |
|---|---|
| I | Less than 35 mm |
| II | From 35 mm to 52 mm |
| III | Over 52 mm |

Note(s): Source: Based on Chow et al., (1988) [1].

The spatiotemporal distribution of rainfall in Colombia presents different antecedent moisture conditions due to its geographical position and according to where floods frequently occur throughout different areas in its extent [8].

The La Niña phenomenon during the 2010–2011 period in the Atlántico region led to human losses, displacements, damage to road infrastructure and material losses in the southern part of the region [8], indicating a need to perform a probabilistic analysis to determine the special distribution of the antecedent moisture that allows engineers and designers to make sound decisions. Therefore, the present study aims to contribute to the following: (a) first, perform the seasonal frequency analysis of the total 5-day antecedent rainfall using four (4) cumulative probability distribution functions (Gev, Gumbel, Pearson Type III and Log Pearson Type III), considering the maximum likelihood, moment method and Sam fit methods; (b) second, to evaluate the IDW, kriging and spline interpolation methods; and (c) finally, to determine the spatial distribution of the antecedent moisture conditions for the Atlántico region for return periods of 2.33, 5, 10, 20, 50 and 100 years.

## 2. Study Area and Data

The study area lies in the coastal zone of the Colombian territory, located in nortwestern South America. The Atlántico region (named as "Atlántico Department" in Colombia) has an area of 3382 km$^2$. It is composed of 22 municipalities and the Special Industrial and Port District of Barranquilla. The coastal zone of the region represents most of the strategic coastal ecosystems of the country: mangroves, soft-bottom communities of the continental shelf, coastal deltas and lagoons, beaches and cliffs. The region is characterized predominantly as a livestock zone [9–12]. Figure 1 shows the political-administrative location of the Atlántico region, where the coordinates are referenced to the Colombian cartographic projection system. The climate of the region is warm and dry, and the average annual temperature is between 28 °C and a maximum of 40 °C. Annual rainfall varies between 500 and 1500 mm.

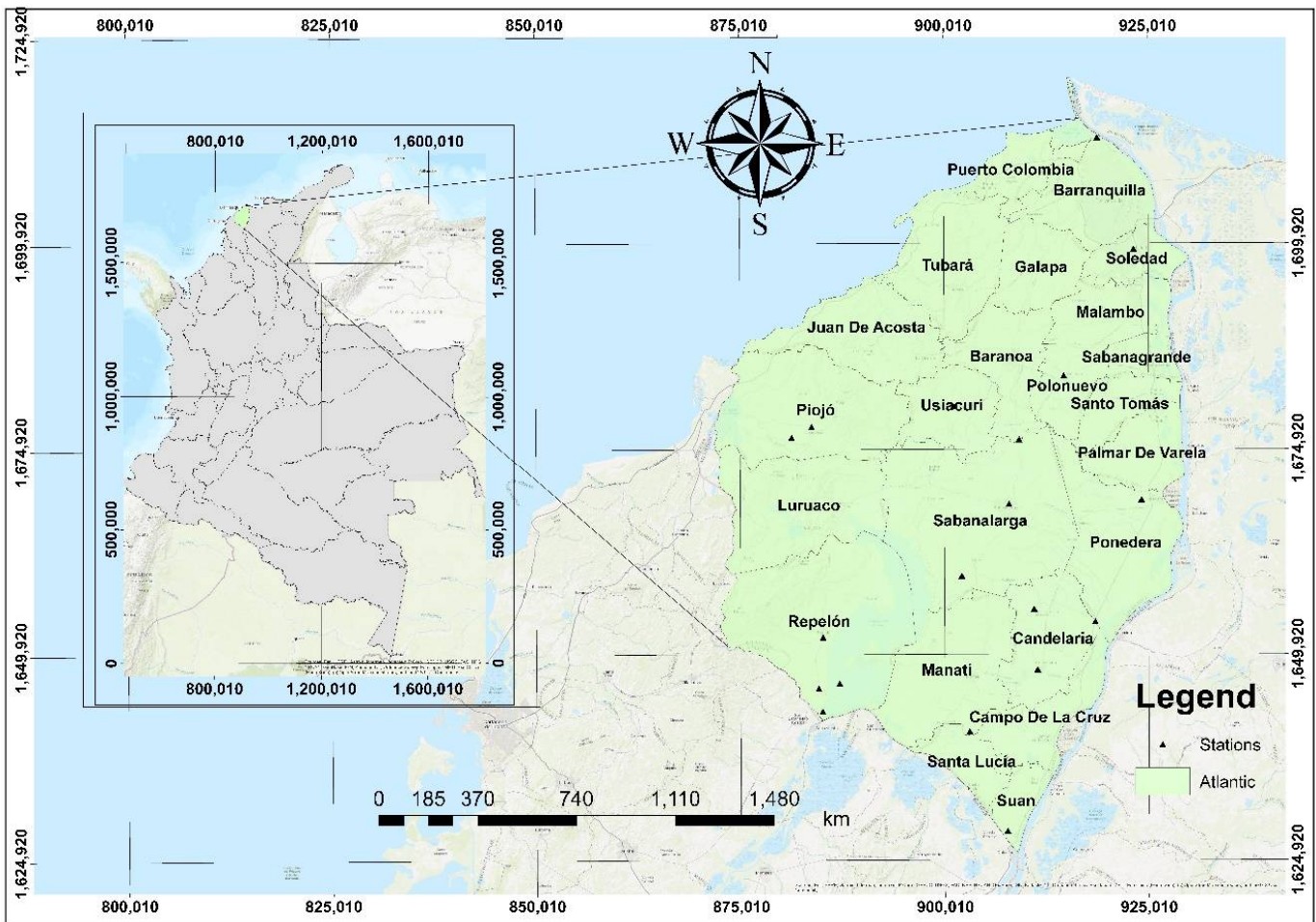

**Figure 1.** Political-administrative boundaries of the Atlántico region.

### 3. Methodology

To obtain the results proposed in the research, the methodology is composed of the following steps.

#### 3.1. Analysis of Data Collection

The analysis of the 24 h maximum rainfall ($P_{daily\text{-}max}$) and of the total antecedent rainfall 5 days prior to the maximum rainfall event was performed considering that the maximum rainfall data over 24 h were collected and provided at the IDEAM (Institute of Hydrology, Meteorology and Environmental Studies) of Colombia. The rainfall station had a minimum of 25 years of observations to ensure reliable results. The used rain gauge stations were: Ernesto Cortissoz Airport (code: 29045020), Candelaria (code: 29040260), Casa de Bombas (code: 29030410), El Porvenir (code: 14010090), Hacienda el Rabón (code: 29040270), Hibaracho (code: 14010020), Las Flores (code: 29045120), Lena (code: 29040200), Loma Grande (code: 29030270), Los Campanos (code: 29040290), Montebello (code: 29040020), Polo Nuevo (code: 29040080), Ponedera (code: 29040070), Puerto Giraldo (code: 29040300), Repelón (code: 29037060), Sabanalarga (code: 29040190), San José (code: 29030140), San Pedrito Alerta (code: 29040310) and Usiacurí (code: 29040240).

For each station, the data recorded from the date of installation to 2015 were analyzed. According to the recommendations of the U.S. Water Resources Council [13] and Cunnane [14], possible outliers were identified and filtered for the 24 h maximum rainfall data. The outliers were not identified for the analysis of the total 5-day antecedent rainfall. The graphs in Figure 2 show a relationship between the years analyzed and the maximum rainfall in 24 h, where it is observed that the Hacienda El Rabón and San José rainfall

stations present a decreasing trend line over time. In 1985 and 1994, these values were eliminated from the stations of Hacienda El Rabón and San José, respectively, since they were outliers.

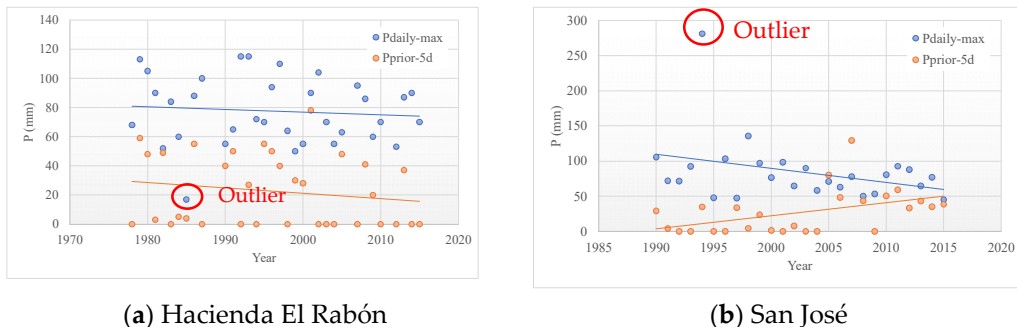

(**a**) Hacienda El Rabón                 (**b**) San José

**Figure 2.** Outlier values of 24 h maximum rainfall records.

The cases in which the maximum rainfall in 24 h occurred several times in the same year were also considered. For example, the Ernesto Cortissoz Airport station 24 h maximum precipitation for 1959 was 55.5 mm, which was repeated in the months of September and October, and the cumulative rainfall obtained during the 5 days prior to extreme annual downpours ($P_{prior\text{-}5d}$) was 0.0 and 10.5 mm, respectively. For the analyses, the maximum value of 10.5 mm was considered.

### 3.2. Seasonal Frequency Analysis

The probability functions for maximum cumulative precipitation allow the frequency of extreme events for different return periods to be analyzed [15]. For the frequency analysis, the data series of maximum rainfall in 24 h or of maximum instantaneous flows can be used [16]. In the current investigation, the Hyfran program was used to perform the frequency analysis of extreme events with the data series of 24 h maximum rainfall ($P_{daily\text{-}max}$) located in the Atlántico region. The Hyfran program (version 1.1) was developed by the National Institute of Scientific Research—Water, Earth and Environment (INRS-ETE) and the Council for Research in the Natural Sciences and Engineering of Canada. This program includes a set of mathematical instruments that allow the statistical analysis of extreme events [16]. Table 2 describes the different distribution functions used in this study [17,18].

### 3.2.1. Akaike Information Criterion (AIC)

The Akaike information criterion (AIC) is an indicator of goodness of fit that allows the comparison of statistical models that differ in complexity and quality of fit. A lower AIC value indicates a better model fit [19]. The criterion is based on information theory and the property of the maximum likelihood method [20]. It is calculated according to Equation (1).

$$\text{AIC} = -2 \left( \begin{array}{c} \text{maximum} \\ \log -\text{likelihood} \\ \text{of model} \end{array} \right) + 2 \left( \begin{array}{c} \text{number of} \\ \text{independent} \\ \text{parameters} \\ \text{of the model} \end{array} \right) \tag{1}$$

**Table 2.** Cumulative distribution functions used in the study.

| Distribution | Cumulative Distribution Function | Range of Random Variables and Parameters |
|---|---|---|
| Gev (3 parameters) | $F(x) = e^{-(1-\frac{k(x-u)}{\alpha})^{1/k}}$ | $u + ^{\alpha}/k \le x \le \infty \ if \ k < 0,$ $-\infty \big\langle x \le u + ^{\alpha}/k \ if \ k \big\rangle 0$ Where : $k$ is the shape parameter, $u$ is the location parameter, and $\alpha$ the scale parameter |
| Gumbel (2 parameters) | $F(x) = \frac{1}{\alpha} exp\left[-\frac{x-u}{\alpha} - exp\left(-\frac{x-u}{\alpha}\right)\right]$ | $-\infty < x < +\infty$ Where : $u$ is the location parameter, and $\alpha$ is the scale parameter |
| Pearson Type III (3 parameters) | $F(x) = \frac{\alpha^{\lambda}}{\Gamma(\lambda)}(x-m)^{\lambda-1}e^{-\alpha(x-m)}$ | Where $\lambda$ is the shape parameter, $\alpha$ is the scale parameter $m$ the location parameter, and $\Gamma(\lambda)$ is the gamma function. |
| Log Pearson Type III (3 parameters) | $F(x) = \frac{\alpha^{\lambda}(y-m)^{\lambda-1}e^{-\alpha(y-m)}}{x\Gamma(\lambda)}$ | $x \ge y \ if \ \gamma > 0; x \le y \ if$ $\alpha < 0, \ y = \ln x \ge m$ $\alpha = \frac{S_y}{\sqrt{\lambda}} , \ \lambda = \left[\frac{2}{C_s(y)}\right]^2$ Where $C_s(y) > 0$ $\varepsilon = \overline{y} - S_y\sqrt{\lambda}$ Where: $\alpha$ is the scale parameter, $\lambda$ the shape parameter, and $m$ the location parameter. |

### 3.2.2. Bayesian Information Criterion (BIC)

The BIC criterion is very similar to the AIC. This more strongly penalizes the probabilistic models with a greater number of estimated parameters; therefore, more inferior models are obtained than those obtained by AIC. This criterion is more prone to overestimating the models [21].

$$\text{BIC} = -2 \begin{pmatrix} \text{maximum} \\ \text{log-likelihood} \\ \text{of model} \end{pmatrix} + \begin{pmatrix} \text{number of} \\ \text{independent} \\ \text{parameters} \\ \text{of the model} \end{pmatrix} \times \begin{pmatrix} \text{Logarithm} \\ \text{of the number} \\ \text{of data points} \end{pmatrix} \quad (2)$$

When comparing several models from a Bayesian approach, a lower BIC value indicates a better model fit; however, this criterion also allows comparison through the conditional probability P(MI|x), which represents the probability that the data are generated by the model. In this sense, the best probabilistic model will be the one with the highest posterior probability [21].

### 3.3. Estimation of Cumulative Precipitation during the 5 Days Prior to the Occurrence of the Annual Maximum Daily Precipitation

For each of the 19 stations with records of total rainfall data accumulated 5 days prior to the occurrence of the extreme downpour, a seasonal frequency analysis was performed using the different probability functions described in the previous Section 3.2 to obtain projections for return periods (RTs) of 2.33, 5, 10, 20, 50 and 100 years. Based on these analyses, the probability functions that fitted best according to the Akaike criterion were determined for each rainfall station.

The return period (RP) is defined as the occurrence of a given rain event, in any particular year, which can be equaled or exceeded by some percentage, and the probability of exceedance (P) is inversely proportional [17] for stationary conditions.

### 3.4. Spatial Distributions of the Type of Antecedent Moisture

For the determination of the spatial distribution of the antecedent moisture conditions, frequency analyses performed for the estimation of the cumulative precipitation during the 5 days prior to the occurrence of the annual maximum daily precipitation were used.

Subsequently, the interpolation methods of IDW, kriging and spline were used to determine the antecedent moisture patterns. Table 3 presents a summary of the applicability of each of these methods.

**Table 3.** Summary of the applicability of the IDW, kriging and spline interpolation methods.

| Interpolation Method | Definition | Background |
|---|---|---|
| Kriging | It is a geostatistical method based on a mathematical formula, taking into account the correlation of the neighboring midpoints to explain surface variations [22]. | It is most commonly used for the analysis of climate variables worldwide [23]. In addition, it has been used in some studies [24,25] for the analysis of the spatial distribution of annual precipitation. |
| IDW | The average inverse distance weighting is one of the most common deterministic methods. It assumes that the influence of the points decreases as the distance between them increases [22]. | According to Vargas et al. [26], this method is the most appropriate for the analysis of rainfall interpolation in the city of Bogotá, Colombia. It has been used for computing the spatial distribution of maximum daily precipitation for various return periods. |
| Spline | It is a deterministic interpolator. It uses a mathematical function to minimize the total curvature of the surface, yielding smooth curves that pass through the input points [22]. | This method has been used in various studies [17,27]. |

The kriging formula is expressed as function of $P_{prior\text{-}5d}$ as

$$P_{prior-5d-RP} = \sum_{i=1}^{N} F_i P_{prior-5d-rfi} \tag{3}$$

where $P_{prior\text{-}5d\text{-}RP}$ is the isohyet line for a return period, $P_{prior\text{-}5d\text{-}rfn}$ is the rainfall value of $P_{prior\text{-}5d}$ calculated for a rainfall station for a return period, N is the number of analyzed rainfall stations, and $F_i$ is an unknown weight for the measured value of the $i$th rainfall station.

The IDW formula to compute an isohyet line of $P_{prior\text{-}5d}$ for a return period was computed using

$$P_{prior-5d-RP} = \frac{W_1 P_{prior-5d-rf1} + W_2 P_{prior-5d-rf2} + \ldots + W_n P_{prior-5d-rfn}}{W_1 + W_2 + \ldots + W_n} \tag{4}$$

where $W_i$ is an adopted weight.

Finally, the spline method uses the following formulation:

$$P_{prior-5d-RP} = T + \sum_{i=1}^{N} \lambda_i R(r_j) \tag{5}$$

where $T$ and $R(r_j)$ depend on the regularized or tension method, and $\lambda_i$ is a coefficient that is computed based on the solution of the system of linear equations.

During the analysis, the methods of kriging, IDW and spline were used based on the default options of ArcGIS.

*3.5. Evaluation of Interpolation Methods*

The prediction accuracy of each interpolation method was evaluated using the root mean square error (RMSE) (Equation (3)). The lower the RMSE values, the better the interpolation method [28].

$$RMSE = \sqrt{\frac{\sum_{i=1}^{n} (P_{simulated} - P_{actual})^2}{n}} \tag{6}$$

where

$P_{actual}$ = Average areal precipitation of the adjusted IDW interpolation method (PIDW-areal).

$P_{simulated}$ = Average areal precipitation of the unadjusted IDW, the kriging and spline (Pm-areal) interpolation methods.

*n*: Dataset of the corresponding scenario

## 4. Results and Discussion

### 4.1. Best Fit Probability Function

Table A1 shows the fit results of the cumulative precipitation during the 5 days prior to the occurrence of the annual maximum daily precipitation for different return periods using the different probability functions described in Table 2. The analysis of the different probability functions shows that the maximum likelihood fit method to establish the cumulative precipitation during the 5 days prior to the occurrence of the annual maximum precipitation in 24 h for different return periods does not converge in most of the cases for the hydrological distributions Gev, Pearson Type II and Log Pearson Type III. The Gumbel probability distribution was the only one that managed to fit the trend of the data for all the records of the rainfall stations. Table A3 shows the parameters determined and the AIC value for each of the stations using the maximum likelihood method.

Similarly, the behavior of the cumulative precipitation during the 5 days prior to the occurrence of the annual maximum daily precipitation was determined using the method of moments. Table A2 shows the estimated values for different return periods, in which it is observed that for all distributions, there is convergence in the estimation. Table A4 shows the fit parameters and the value of the AIC test criterion.

Table A5 shows the consolidation to establish the best probability distribution to fit the cumulative precipitation during the 5 days prior to the occurrence of the annual maximum daily precipitation using the AIC. Table 4 shows that Pearson Type III at the regional scale is the distribution function that best fits the rainfall data analyzed in the study with 52.63%, followed by Gumbel with 47.37%, while Gev did not have the best fit in any of the cases.

**Table 4.** Best fit distribution function.

| Distribution Function | No. Times of Best Fit (AIC) | Percentage (%) of Best Fit |
|---|---|---|
| Gev | 0 | 0 |
| Gumbel | 9 | 47.37 |
| Pearson Type III | 10 | 52.63 |
| Pearson Log Type III | No estimate was obtained | 0 |
| Total | 19 | 100 |

### 4.2. Evaluation of the Spatial Distribution of Cumulative Rainfall during the 5 Days

Once the visual inspection of the spatial distribution of rainfall was performed, it was observed that, among all methods, the IDW presented fewer inconsistencies; however, it was necessary to perform a manual fit in some areas. Figure 3 shows the results obtained by the different methods. The spline method generated spatial distributions with negative values based on a return period of 100 years. The kriging interpolation method showed areas with little isoline interpolation. The IDW method, although some adjustments were made, did not present this type of inconsistency.

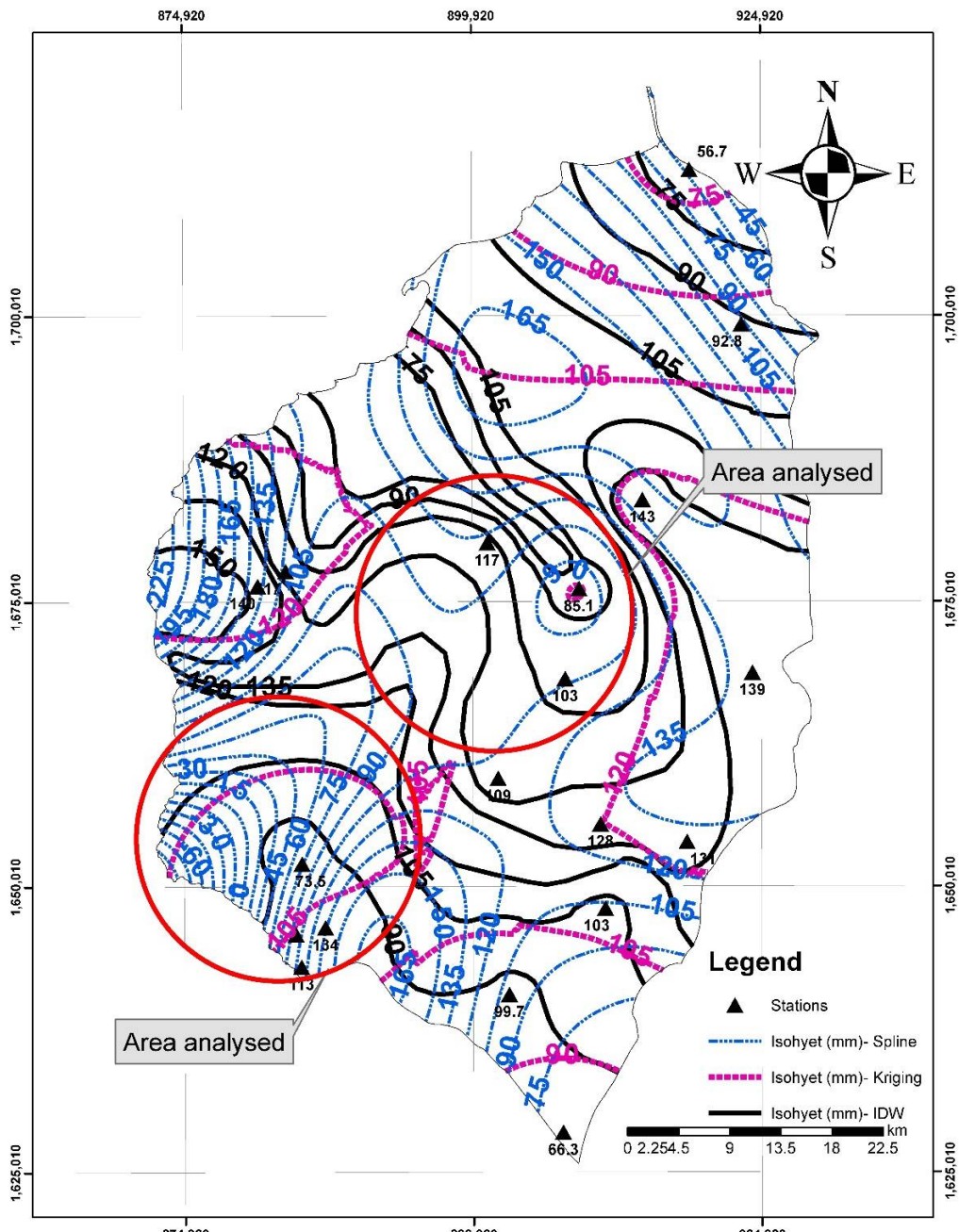

**Figure 3.** Comparison of spatial interpolation methods. Spatial distribution of $P_{prior\text{-}5d}$ for a 100-year RP by using spline, kriging and IDW method.

In addition to visual inspection, the spatial distribution of rainfall was evaluated in three drainage basins of different sizes located at different distances from the nearby rainfall stations. Drainage basin 1 (C1) is located between the Luruaco and Repelón municipalities, drainage basin 2 (C2) in the Malambo municipality and drainage basin 3 (C3) in the Sabanalarga municipality (see Figure 4). Table 5 presents the summary of the information on the drainage basins and their nearest rain gauges.

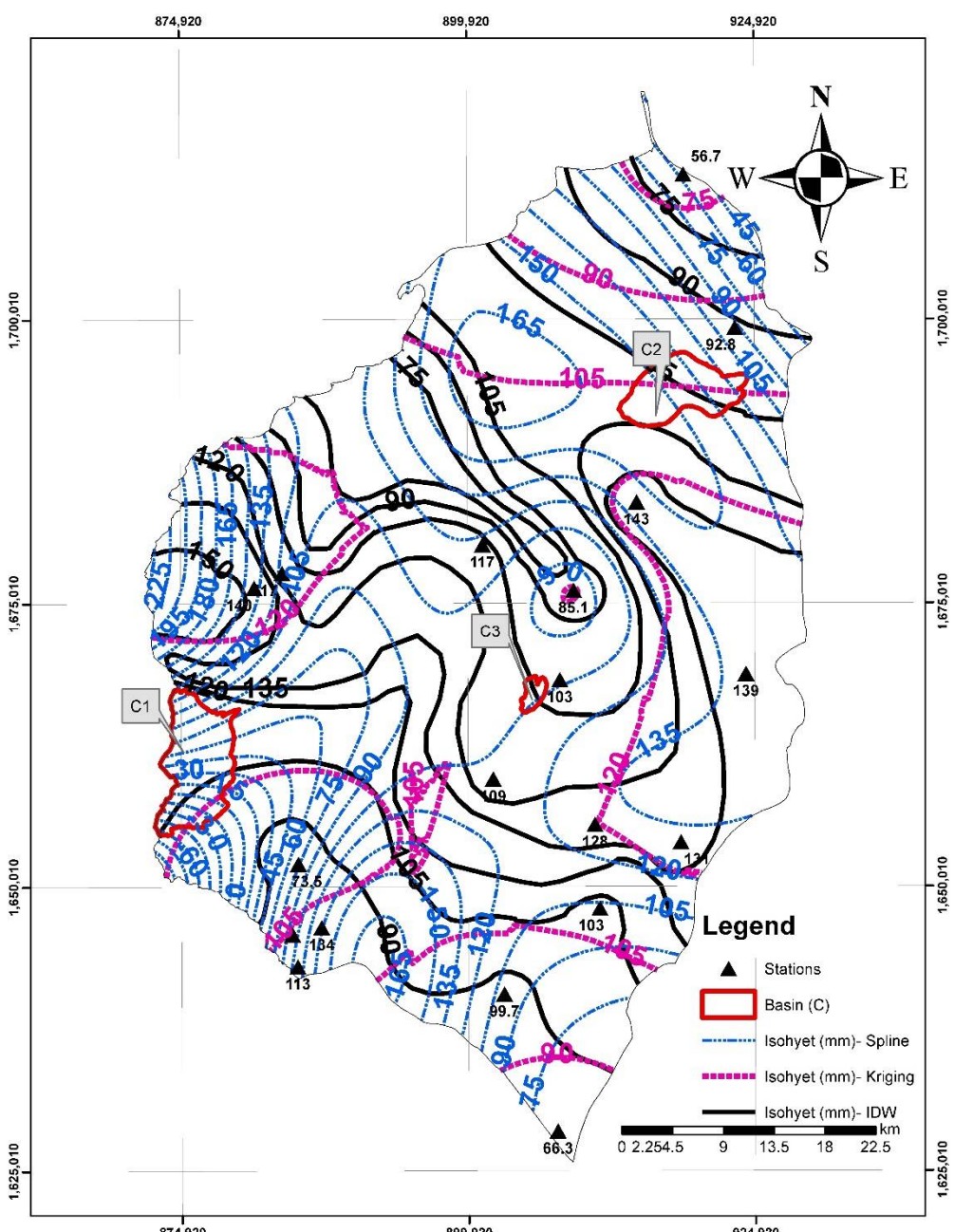

**Figure 4.** Determination of antecedent moisture for drainage basins C1, C2 and C3. Spatial distribution of precipitation by using spline, kriging, and IDW method.

**Table 5.** Summary of drainage basin information (C).

| Drainage Basin | Area (ha) | Nearest Rainfall Station | Distance from the Rainfall Station to the Basin Centroid (km) |
|---|---|---|---|
| C1 | 5851.87 | Repelón | 12.49 |
| C2 | 448.72 | Apto Ernesto Cortizo | 6.43 |
| C3 | 4316.32 | Sabanalarga | 2.6 |

Taking into account the different interpolation methods, the average areal precipitation of basins C1, C2 and C3 was estimated for return periods of 2.33, 5, 10, 20, 50 and 100 years.

Table 6 shows the results of average cumulative precipitation during the 5 days prior to the extreme annual areal storm that were obtained for each of the drainage basins.

**Table 6.** Values of PIDW-areal and Pm-areal for the different interpolation methods.

| Interpolation Method | Drainage Basin | $P_{IDW-areal}$ (mm) | | | | | |
|---|---|---|---|---|---|---|---|
| | | RP (Years) | | | | | |
| | | 2.33 | 5 | 10 | 20 | 50 | 100 |
| Adjusted IDW | C1 | 22.50 | 43.47 | 61.76 | 75.44 | 94.91 | 108.68 |
| | C2 | 19.68 | 37.75 | 53.47 | 69.92 | 90.32 | 105.47 |
| | C3 | 17.50 | 38.72 | 55.00 | 72.08 | 90.40 | 106.36 |
| | | $P_{m-areal}$ (mm) | | | | | |
| IDW | C1 | 23.90 | 43.47 | 58.07 | 74.50 | 94.74 | 108.47 |
| | | $P_{m-areal}$ (mm) | | | | | |
| IDW | C2 | 19.68 | 37.75 | 54.36 | 69.89 | 90.46 | 105.93 |
| | C3 | 17.50 | 37.91 | 55.00 | 72.09 | 91.29 | 106.44 |
| Kriging | C1 | 24.59 | 42.52 | 58.28 | 75.00 | 97.50 | 112.50 |
| | C2 | 18.77 | 38.66 | 55.00 | 69.42 | 90.45 | 106.16 |
| | C3 | 22.50 | 39.23 | 55.00 | 75.00 | 97.50 | 112.50 |
| Spline | C1 | 20.29 | 27.07 | 30.93 | 34.90 | 41.50 | 45.68 |
| | C2 | 21.20 | 46.18 | 69.12 | 91.80 | 119.19 | 142.40 |
| | C3 | 17.50 | 37.50 | 55.00 | 67.58 | 84.88 | 99.56 |

From the cumulative precipitation during the 5 areal days, the root mean square error (RMSE) was estimated. Table 7 shows the RMSE results obtained for the different interpolation methods. The maximum and minimum RMSE values obtained by the IDW method were 2.19 and 0.33 mm, respectively. These results confirm that the manual adjustments made to this method were minimal. The RMSE values of the kriging method range between 0.81 and 4.37 mm. This method also did not show many variations with respect to the adjusted IDW method. The variations in the kriging method with respect to the adjusted IDW occurred due to the differences in average areal precipitation in the C3 basin, located exactly in the area where kriging presented little isoline interpolation. The RMSE results obtained by the spline method range between 1.55 and 42.35 mm. The large discrepancy of the spline method with respect to the adjusted IDW is because spatial rainfall distributions with negative values were generated over basin C1. This caused the underestimation of the mean areal precipitation in the C1 basin and, therefore, a higher mean square error.

**Table 7.** Evaluation of interpolation methods in the different basins (C).

| Interpolation Method | Drainage Basin | Root Mean Square Error—RMSE | | | | | |
|---|---|---|---|---|---|---|---|
| | | RP (Years) | | | | | |
| | | 2.33 | 5 | 10 | 20 | 50 | 100 |
| IDW | | 0.81 | 0.47 | 2.19 | 0.55 | 0.53 | 0.30 |
| Kriging | | 3.17 | 0.81 | 2.19 | 1.73 | 4.37 | 4.19 |
| Spline | | 1.55 | 10.67 | 14.51 | 26.72 | 35.19 | 42.35 |

*4.3. Spatial Distribution of the Antecedent Moisture Conditions for the Atlántico Region for Return Periods 2.33, 5, 10, 20, 50 and 100 Years*

Figures 5–10 show the spatial variation of the antecedent moisture conditions for the different return periods. The spatial variation of the type of antecedent moisture ranges between 15 and 30 mm for a return period of 2.33 years (Figure 5) and between 25 and

60 mm for a return period of 5 years (Figure 6), between 35 and 80 mm for an RP of 10 years (Figure 7), between 40 and 90 mm for 20 years (Figure 8), between 60 and 135 mm for a period of 50 years (Figure 9) and between 75 and 150 mm for a 100-year return period (Figure 10). In addition, it is noteworthy that for a return period of 2.33 years, it was found that the entire Atlántico area is in antecedent moisture condition AMC I. For a 5-year return period, antecedent moisture conditions AMC I, II and III are observed. For the return periods 10, 20 and 50 years, only zones in AMC II and III conditions are identified, while for 100 years, the zones are entirely in AMC III condition. Table 8 shows a summary of the spatial distributions of antecedent moisture, area and percentage thereof.

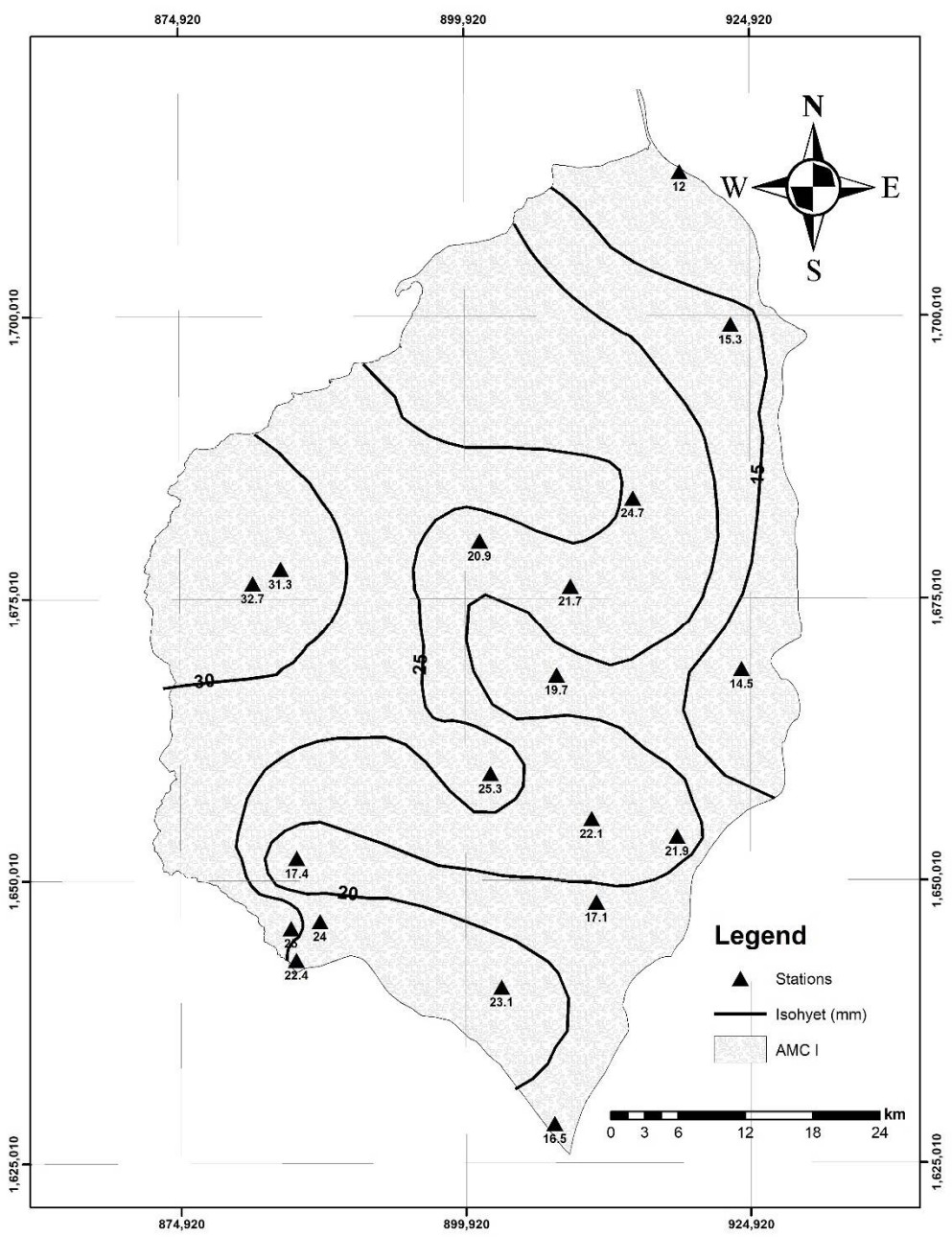

**Figure 5.** Spatial distribution of $P_{prior\text{-}5d}$ of the type of moisture conditions for a return period of 2.33 years.

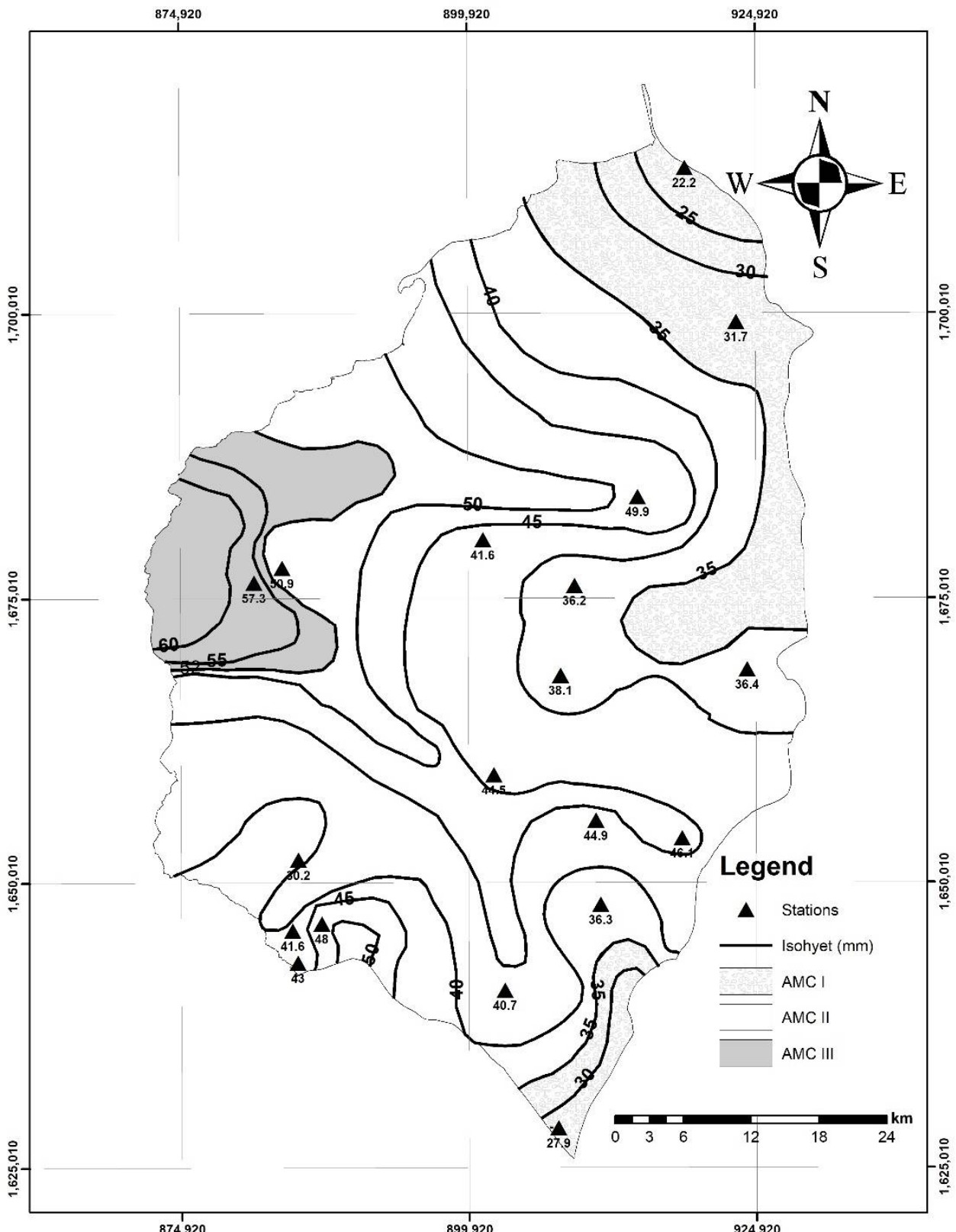

**Figure 6.** Spatial distribution of $P_{prior-5d}$ of the type of moisture conditions for a return period of 5 years.

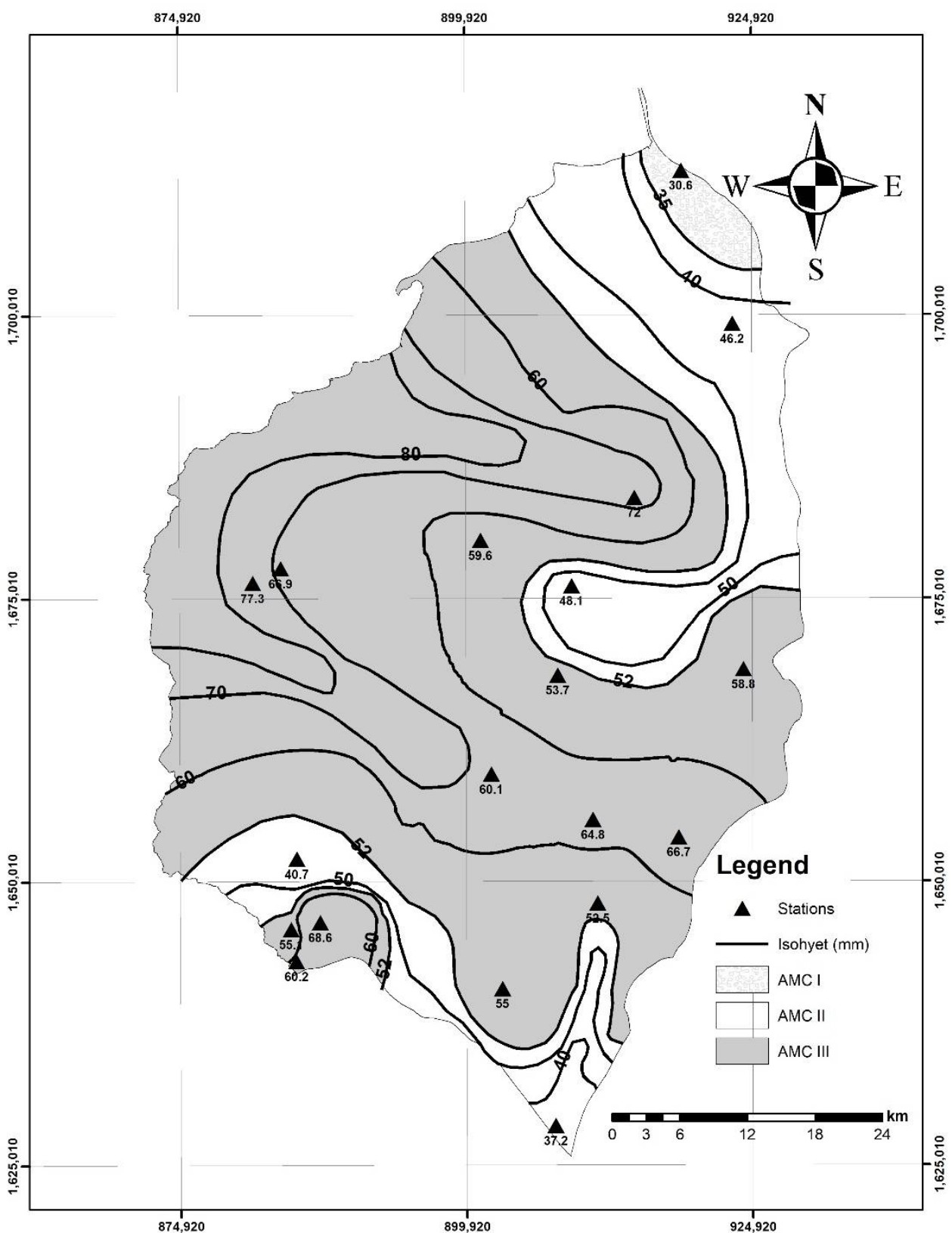

**Figure 7.** Spatial distribution of $P_{prior\text{-}5d}$ of the type of moisture conditions for a return period of 10 years.

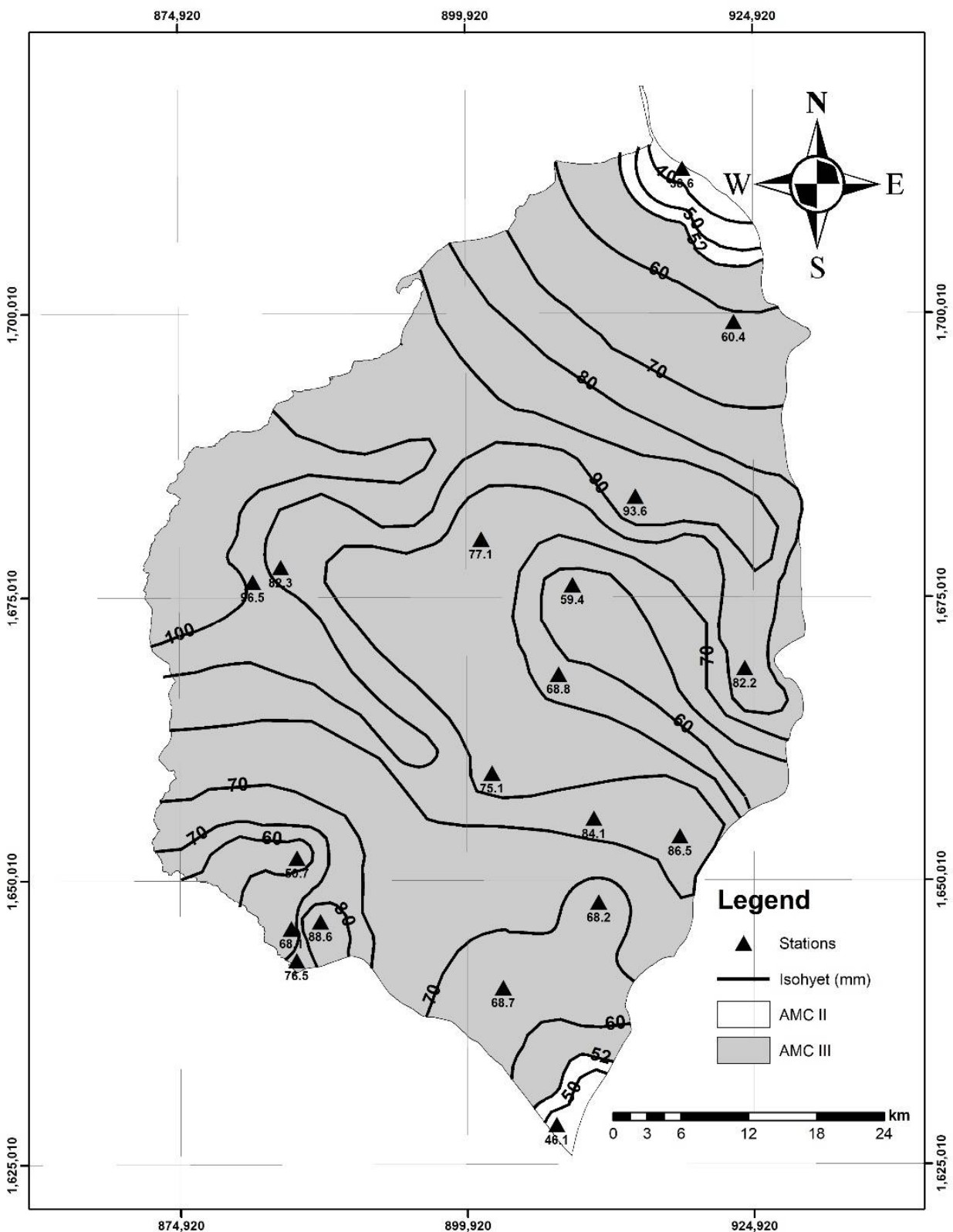

**Figure 8.** Spatial distribution of $P_{prior\text{-}5d}$ of the type of moisture conditions for a return period of 20 years.

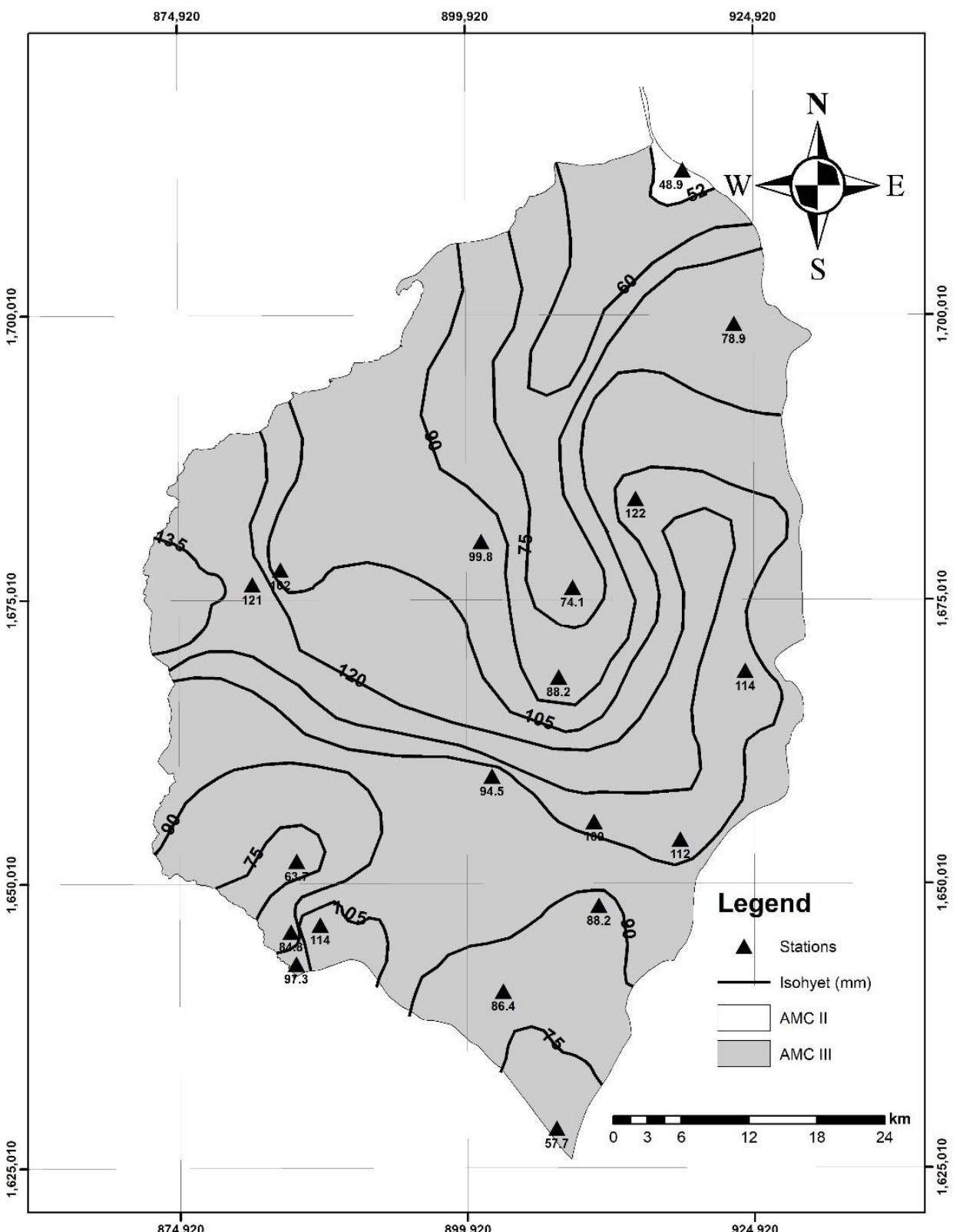

**Figure 9.** Spatial distribution of $P_{prior-5d}$ of the type of moisture conditions for a return period of 50 years.

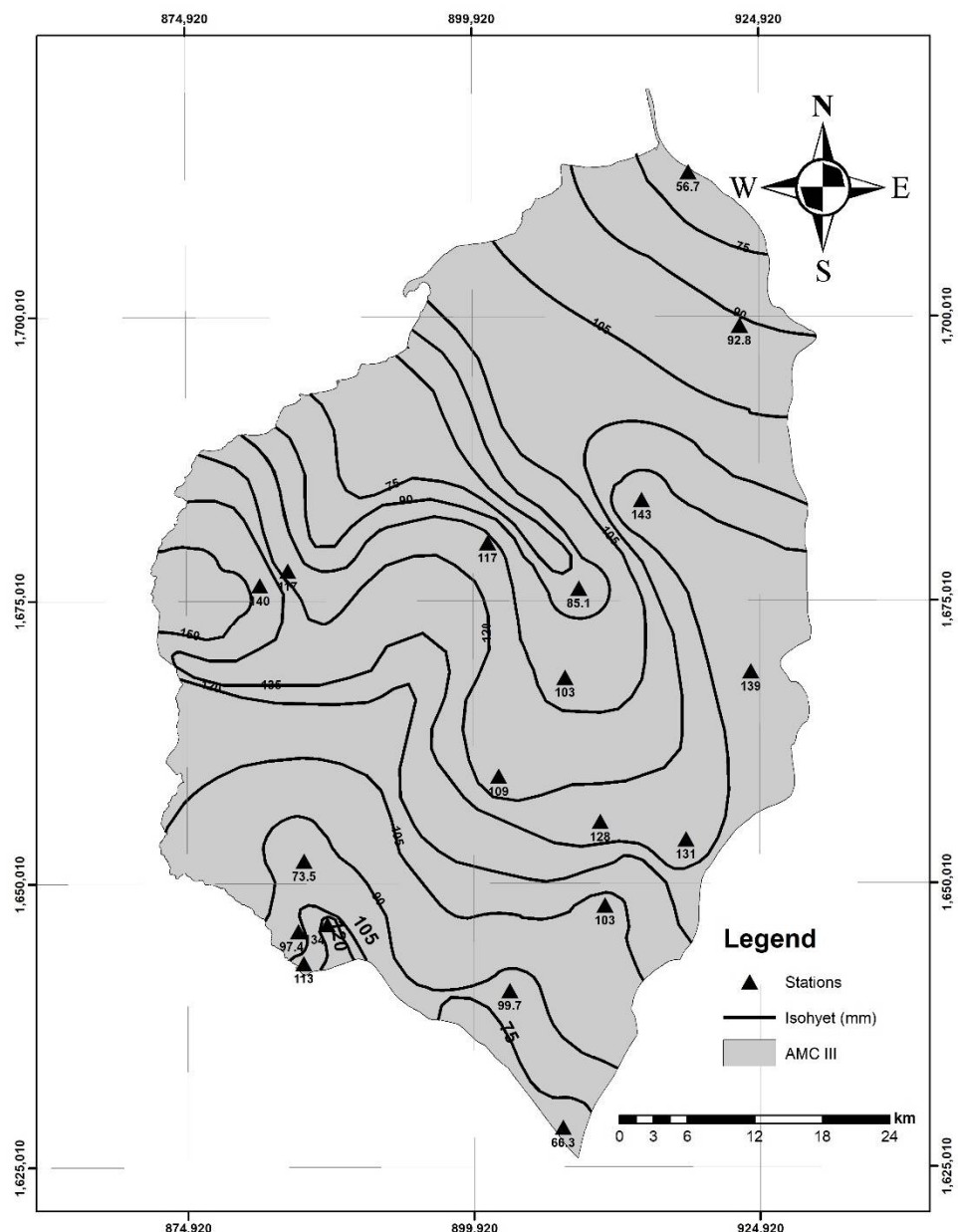

**Figure 10.** Spatial distribution of $P_{prior-5d}$ of the type of moisture conditions for a return period of 100 years.

**Table 8.** Areas of moisture conditions AMC I, AMC II and AMC III.

| Return Period (RP) | AMC I | | AMC II | | AMC III | | Range of Spatial Distribution (mm) |
|---|---|---|---|---|---|---|---|
| | Area (km²) | Percentage (%) | Area (km²) | Percentage (%) | Area (km²) | Percentage (%) | |
| 2.33 | 3313.00 | 100.00 | 0.00 | 0.00 | 0.00 | 0.00 | 15–30 |
| 5 | 505.70 | 15.26 | 2523.81 | 76.18 | 283.49 | 8.50 | 30–60 |
| 10 | 54.52 | 1.65 | 828.57 | 25.01 | 2429.16 | 73.32 | 40–80 |
| 20 | 0.00 | 0.00 | 92.75 | 2.80 | 3219.50 | 97.18 | 50–110 |
| 50 | 0.00 | 0.00 | 64.48 | 1.95 | 3247.77 | 98.03 | 75–135 |
| 100 | 0.00 | 0.00 | 0.00 | 0.00 | 3313.00 | 100.00 | 75–150 |

The spatial distribution of soil moisture content has been studied by different authors [29–31]. The spatial distribution of moisture can be variable even for small watersheds

and should not be assumed to be constant because this could lead to modeling problems. This is evidenced in Figure 11, which shows the variations in the antecedent moisture conditions present in the drainage basins (C). For a 5-year return period, the C2 basin shows a spatial distribution of antecedent moisture content classified as AMC I and AMC II. This spatial distribution for the same basin for a 10-year return period presents AMC II and III conditions. For a return period of 100 years, the spatial distribution of the moisture content shows a classification of AMC III. On the other hand, drainage basins C1 and C3 present a spatial distribution consistent with AMC II for a return period of 5 years and AMC III for return periods of 10 and 100 years.

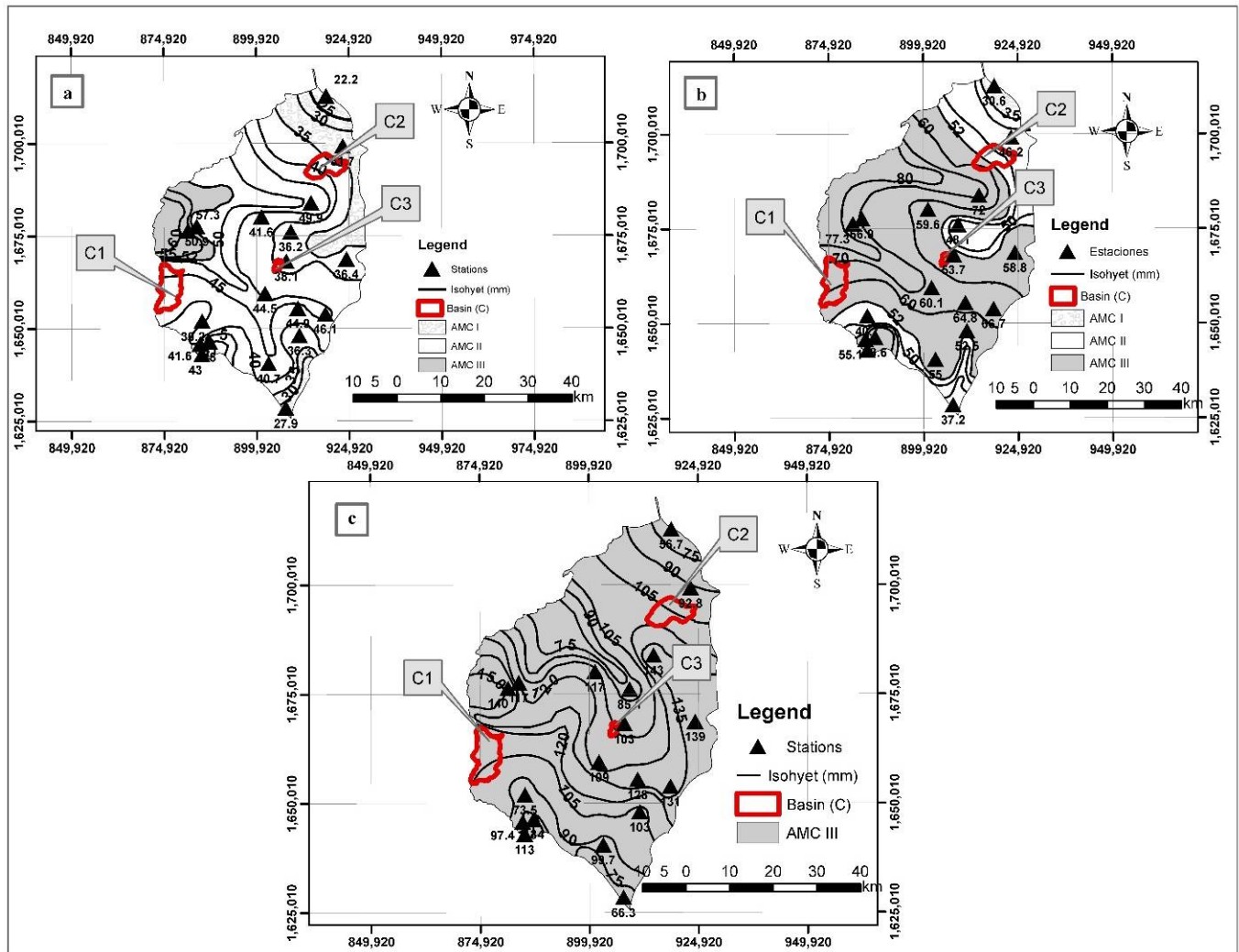

**Figure 11.** Spatial distribution of $P_{prior-5d}$ of the antecedent moisture content for basin C1, C2 and C3. Spatial distribution of the type of moisture in the basins considering: (**a**) a return period of 5 years; (**b**) a 10-year RP; (**c**) 100-year RP.

It is very important to consider that the maximum retention ($S_{RP}$) can vary according to a defined return period, as well as the curve number ($CN_{RP}$), since the antecedent moisture conditions vary for a specific return period. The weighted average of the curve number ($\overline{CN_{RP}}$) should be computed for each drainage basin. In this sense, the formula to describe the maximum retention should be described as

$$S_{RP} = \frac{25400}{CN_{RP}} - 254 \qquad (7)$$

Bearing this in mind, designers and engineers should address the proposed methodology in this research in actual basins to compute the maximum retention and the total water flow for a suitable estimation associated with various return periods.

## 5. Conclusions

In the development of this study, the spatial variation of the antecedent moisture conditions for the Atlántico region was determined for different return periods based on a proposed probabilistic approach, which is composed of several steps: (i) analysis of data collection; (ii) a seasonal frequency analysis, including the application of the Akaike and Bayesian Information Criteria and the estimation of cumulative precipitation during the 5 days prior to the annual maximum daily precipitation; and (iii) the computation of spatial distributions of the antecedent moisture conditions applying difference interpolation methods.

The probabilistic approach was applied to the Atlántico region in Colombia. For the analyzed case study, the seasonal frequency analysis of the total 5-day antecedent precipitation was performed using four (4) cumulative probability distribution functions (Gev, Gumbel, Pearson Type III and Log Pearson Type III), considering the maximum likelihood, method of moments and Sam fit methods. The results indicated that Pearson Type III at the regional scale was the distribution function that best fitted the rainfall data at 52.63% of the stations analyzed, followed by Gumbel at 47.37%. The interpolation methods of IDW, kriging and spline were evaluated in three (3) basins of different sizes. The results show that the IDW interpolation method presents better results for the analysis of the spatial distribution of antecedent moisture. The kriging interpolation method showed little isoline interpolation. The root mean square error (RMSE) showed that in these areas, precipitation can be over-estimated. The spline interpolation method tends to underestimate the model due to the spatial distribution of moisture content with negative values.

It is important to consider that the proposed analysis can help engineers and designers compute the antecedent moisture conditions, which can be used to compute the curve number for a return period.

The frequency analysis of cumulative rainfall obtained during the 5 days prior to extreme annual downpour was conducted using stationary conditions. Future works should involve considering non-stationary frequency analysis.

**Author Contributions:** Conceptualization, J.J.S.-C. and O.E.C.-H.; methodology, J.R.C.-H.; formal analysis, G.G. and R.L.; writing—original draft preparation, J.J.S.-C. and O.E.C.-H.; writing—review and editing, J.R.C.-H.; supervision, G.G. and R.L. All authors have read and agreed to the published version of the manuscript.

**Funding:** This research was funded by University of Bío-Bío grant number 2060222 IF/R, 2060240 IF/R and 2160277 GI/EF, and Universidad Andres Bello grant number DI-12-20/REG.

**Institutional Review Board Statement:** Not applicable.

**Informed Consent Statement:** Not applicable.

**Data Availability Statement:** The data presented in this study are available on request from the corresponding author.

**Acknowledgments:** This research was supported by University of Bío-Bío grant number 2060222 IF/R, 2060240 IF/R and 2160277 GI/EF, and Universidad Andres Bello grant number DI-12-20/REG.

**Conflicts of Interest:** The authors declare no conflict of interest.

# Appendix A

**Table A1.** Fit results of the cumulative precipitation during the 5 days prior to the occurrence of the annual maximum daily precipitation for different return periods using the different probability functions and maximum likelihood fit.

| | Cumulative Precipitation during the 5 Days Prior to the Occurrence of the Annual Maximum Daily Precipitation (mm) | | | | | | | | | | | | | | | | | | | | | | | |
| --- | --- | --- | --- | --- | --- | --- | --- | --- | --- | --- | --- | --- | --- | --- | --- | --- | --- | --- | --- | --- | --- | --- | --- | --- |
| | Gev | | | | | | Gumbel | | | | | | Pearson Type III | | | | | | Log Pearson Type III | | | | | |
| | RP (Years) | | | | | | RP (Years) | | | | | | RP (Years) | | | | | | RP (Years) | | | | | |
| Rainfall Station | 2.33 | 5 | 10 | 20 | 50 | 100 | 2.33 | 5 | 10 | 20 | 25 | 100 | 2.33 | 5 | 10 | 20 | 50 | 100 | 2.33 | 5 | 10 | 20 | 50 | 100 |
| Aeropuerto Ernesto Cortissoz | NC | NC | NC | NC | NC | NC | 16.5 | 28.5 | 38.4 | 47.8 | 60 | 69.1 | NC | NC | NC | NC | NC | NC | NC | NC | NC | NC | NC | NC |
| Candelaria | NC | NC | NC | NC | NC | NC | 16.6 | 31.3 | 43.2 | 54.6 | 69.4 | 80.5 | NC | NC | NC | NC | NC | NC | NC | NC | NC | NC | NC | NC |
| Casa de Bombas | NC | NC | NC | NC | NC | NC | 22.3 | 38.9 | 52.5 | 65.5 | 82.3 | 94.9 | NC | NC | NC | NC | NC | NC | NC | NC | NC | NC | NC | NC |
| El Porvenir | NC | NC | NC | NC | NC | NC | 32.7 | 57.3 | 77.3 | 96.5 | 121 | 140 | NC | NC | NC | NC | NC | NC | NC | NC | NC | NC | NC | NC |
| Hacienda el Rabón | NC | NC | NC | NC | NC | NC | 22.6 | 40.2 | 54.6 | 68.4 | 86.2 | 99.6 | NC | NC | NC | NC | NC | NC | NC | NC | NC | NC | NC | NC |
| Hibaracho | 28.4 | 49.7 | 69.9 | 92 | 125 | 154 | 30.8 | 50.5 | 66.5 | 81.9 | 102 | 117 | NC | NC | NC | NC | NC | NC | NC | NC | NC | NC | NC | NC |
| Las Flores | NC | NC | NC | NC | NC | NC | 12 | 22.2 | 30.6 | 38.6 | 48.9 | 56.7 | NC | NC | NC | NC | NC | NC | NC | NC | NC | NC | NC | NC |
| Lena | NC | NC | NC | NC | NC | NC | 23.5 | 40.7 | 54.7 | 68.1 | 85.5 | 98.6 | NC | NC | NC | NC | NC | NC | NC | NC | NC | NC | NC | NC |
| Loma Grande | 19.7 | 40.6 | 66.3 | 102 | 171 | 249 | 25 | 41.6 | 55.1 | 68.1 | 84.8 | 97.4 | NC | NC | NC | NC | NC | NC | NC | NC | NC | NC | NC | NC |
| Los Campanos | NC | NC | NC | NC | NC | NC | 25.3 | 44.5 | 60.1 | 75.1 | 94.5 | 109 | NC | NC | NC | NC | NC | NC | NC | NC | NC | NC | NC | NC |
| Montebello | 14.2 | 37 | 74.4 | 141 | 317 | 578 | 21.4 | 36 | 47.9 | 59.3 | 74.1 | 85.2 | NC | NC | NC | NC | NC | NC | NC | NC | NC | NC | NC | NC |
| Polo Nuevo | NC | NC | NC | NC | NC | NC | 26.7 | 46.6 | 62.9 | 78.4 | 98.6 | 114 | NC | NC | NC | NC | NC | NC | NC | NC | NC | NC | NC | NC |
| Ponedera | NC | NC | NC | NC | NC | NC | 19.7 | 35 | 47.4 | 59.4 | 74.8 | 86.4 | NC | NC | NC | NC | NC | NC | NC | NC | NC | NC | NC | NC |
| Puerto Giraldo | NC | NC | NC | NC | NC | NC | 22.1 | 39.9 | 54.4 | 68.3 | 86.3 | 99.8 | NC | NC | NC | NC | NC | NC | NC | NC | NC | NC | NC | NC |
| Repelón | 7.35 | 33.7 | 112 | 351 | 1540 | 4660 | 17.4 | 30.2 | 40.7 | 50.7 | 63.7 | 73.5 | NC | NC | NC | NC | NC | NC | NC | NC | NC | NC | NC | NC |
| Sabanalarga | NC | NC | NC | NC | NC | NC | 20.3 | 34.5 | 46.1 | 57.1 | 71.5 | 82.2 | NC | NC | NC | NC | NC | NC | NC | NC | NC | NC | NC | NC |
| San Jose | NC | NC | NC | NC | NC | NC | 25.5 | 44.9 | 60.7 | 75.9 | 95.5 | 110 | NC | NC | NC | NC | NC | NC | NC | NC | NC | NC | NC | NC |
| San Pedrito Alerta | NC | NC | NC | NC | NC | NC | 16.5 | 27.9 | 37.2 | 46.1 | 57.7 | 66.3 | NC | NC | NC | NC | NC | NC | NC | NC | NC | NC | NC | NC |
| Usiacurí | NC | NC | NC | NC | NC | NC | 21.8 | 37.6 | 50.6 | 62.9 | 79 | 91 | NC | NC | NC | NC | NC | NC | NC | NC | NC | NC | NC | NC |

Note(s): The NC cells indicate that it was not possible to obtain an estimate using the maximum likelihood (ML) fit.

**Table A2.** Fit results of the cumulative precipitation during the 5 days prior to the occurrence of the annual maximum daily precipitation for different return periods using the different probability functions and fit of the moment method.

| | Cumulative Precipitation during the 5 Days Prior to the Occurrence of the Annual Maximum Daily Precipitation (mm) | | | | | | | | | | | | | | | | | |
|---|---|---|---|---|---|---|---|---|---|---|---|---|---|---|---|---|---|---|
| | Gev | | | | | | Gumbel | | | | | | Pearson Type III | | | | | |
| | RP (Years) | | | | | | RP (Years) | | | | | | RP (Years) | | | | | |
| **Rainfall Station** | **2.33** | **5** | **10** | **20** | **50** | **100** | **2.33** | **5** | **10** | **20** | **25** | **100** | **2.33** | **5** | **10** | **20** | **50** | **100** |
| Aeropuerto Ernesto Cortissoz | 16.6 | 31.5 | 44.6 | 57.9 | 76.4 | 91.3 | 17.7 | 33.2 | 45.9 | 58 | 73.6 | 85.4 | 15.3 | 31.7 | 46.2 | 60.4 | 78.9 | 92.8 |
| Candelaria | 17.3 | 35.6 | 51.3 | 67.1 | 88.6 | 106 | 18.3 | 37.1 | 52.3 | 67 | 85.9 | 100.0 | 16.2 | 36 | 53 | 69.4 | 90.6 | 106 |
| Casa de Bombas | 23 | 42.5 | 58.8 | 74.8 | 96.2 | 113 | 23.6 | 43.3 | 59.3 | 74.7 | 94.6 | 109.0 | 22.4 | 43 | 60.2 | 76.5 | 97.3 | 113 |
| El Porvenir | 34.1 | 63.2 | 87.5 | 111 | 143 | 167 | 34.9 | 64.3 | 88.2 | 111 | 141.0 | 163.0 | 33.3 | 64.1 | 89.6 | 114 | 144 | 167 |
| Hacienda el Rabón | 25.5 | 43.1 | 55.8 | 66.7 | 79.2 | 87.5 | 23.1 | 40.7 | 55 | 68.7 | 86.4 | 99.7 | 25.8 | 43 | 55.3 | 66.1 | 78.9 | 87.8 |
| Hibaracho | 32.9 | 52.8 | 67.8 | 81.2 | 97.2 | 108 | 31.3 | 50.9 | 66.9 | 82.3 | 102 | 117 | 33.2 | 53 | 67.7 | 80.8 | 96.7 | 108 |
| Las Flores | 12.9 | 25.8 | 36.4 | 46.9 | 60.7 | 71.3 | 13.2 | 26.2 | 36.7 | 46.8 | 59.9 | 69.7 | 12.6 | 26.1 | 37.3 | 47.9 | 61.3 | 71.1 |
| Lena | 23.7 | 44.5 | 62.6 | 81 | 106 | 127 | 25.1 | 46.6 | 64.2 | 81 | 103 | 119 | 22.1 | 44.9 | 64.8 | 84.1 | 109 | 128 |
| Loma Grande | 25.8 | 43.7 | 58.2 | 72.1 | 90 | 103 | 25.8 | 43.6 | 58.2 | 72.1 | 90.2 | 104 | 25.6 | 44.2 | 59.1 | 72.9 | 90.2 | 103 |
| Los Campanos | 26.6 | 49.6 | 68.7 | 87.4 | 112 | 131 | 27.1 | 50.3 | 69.1 | 87.3 | 111 | 128 | 25.9 | 50.3 | 70.3 | 89.2 | 113 | 131 |
| Montebello | 23.1 | 37.8 | 48.7 | 58.4 | 69.9 | 77.9 | 21.7 | 36.2 | 48.1 | 59.4 | 74.1 | 85.1 | 23.3 | 37.8 | 48.6 | 58.1 | 69.5 | 77.6 |
| Polo Nuevo | 26.6 | 49.8 | 69.9 | 90.3 | 119 | 141 | 28.2 | 52.2 | 71.6 | 90.3 | 115 | 133 | 24.9 | 50.2 | 72.3 | 93.9 | 122 | 143 |
| Ponedera | 18.6 | 38.4 | 56.7 | 76.4 | 105 | 130 | 21.4 | 43.1 | 60.8 | 77.7 | 99.7 | 116 | 14.5 | 36.4 | 58.8 | 82.2 | 114 | 139 |
| Puerto Giraldo | 23.1 | 45.5 | 64.7 | 83.1 | 110 | 130 | 24.2 | 47.1 | 65.8 | 83.7 | 107 | 124 | 21.9 | 46.1 | 66.7 | 86.5 | 112 | 131 |
| Repelón | 18.3 | 33.2 | 45.3 | 57 | 72.1 | 83.5 | 18.4 | 33.3 | 45.4 | 57 | 72 | 83.3 | 18.1 | 33.6 | 46.1 | 57.8 | 72.4 | 83 |
| Sabanalarga | 20.7 | 37.6 | 52.1 | 66.7 | 86.4 | 102 | 21.6 | 38.9 | 53 | 66.6 | 84.1 | 97.3 | 19.7 | 38.1 | 53.7 | 68.8 | 88.2 | 103 |
| San Jose | 25.3 | 47.5 | 66.5 | 85.7 | 112 | 133 | 26.6 | 49.3 | 67.8 | 85.6 | 109 | 126 | 24 | 48 | 68.6 | 88.6 | 114 | 134 |
| San Pedrito Alerta | 17.3 | 29.9 | 39.8 | 49.3 | 61.3 | 70.1 | 17.1 | 29.5 | 39.7 | 49.4 | 62 | 71.4 | 17.3 | 30.2 | 40.3 | 49.6 | 61.2 | 69.6 |
| Usiacurí | 22.2 | 41.2 | 57.7 | 74.3 | 97.3 | 116 | 23.4 | 43 | 59 | 74.3 | 94.1 | 109 | 20.9 | 41.6 | 59.6 | 77.1 | 99.8 | 117 |

**Table A3.** Summary of parameters and AIC values of the different distribution functions using the maximum likelihood fit.

| Rainfall Station | Gev | | | | Gumbel | | | Pearson Type III | | | | Log Pearson Type III | | | |
|---|---|---|---|---|---|---|---|---|---|---|---|---|---|---|---|
| | $\alpha$ | k | u | AIC | $\alpha$ | u | AIC | $\alpha$ | $\lambda$ | m | AIC | $\alpha$ | $\lambda$ | m | AIC |
| Aeropuerto Ernesto Cortissoz | NC | NC | NC | NC | 13.08 | 8.91 | 623.33 | NC | NC | NC | NC | NC | NC | NC | NC |
| Candelaria | NC | NC | NC | NC | 19.75 | 7.35 | 287.022 | NC | NC | NC | NC | NC | NC | NC | NC |
| Casa de Bombas | NC | NC | NC | NC | 18.07 | 11.8 | 302.75 | NC | NC | NC | NC | NC | NC | NC | NC |
| El Porvenir | NC | NC | NC | NC | 26.69 | 17.26 | 279.43 | NC | NC | NC | NC | NC | NC | NC | NC |
| Hacienda el Rabón | NC | NC | NC | NC | 18.73 | 11.07 | 303.57 | NC | NC | NC | NC | NC | NC | NC | NC |
| Hibaracho | 19.17 | −0.18 | 16.69 | NC | 21.35 | 18.47 | 421.97 | NC | NC | NC | NC | NC | NC | NC | NC |
| Las Flores | NC | NC | NC | NC | 11.12 | 5.53 | 249.02 | NC | NC | NC | NC | NC | NC | NC | NC |
| Lena | NC | NC | NC | NC | 18.67 | 12.7 | 434.01 | NC | NC | NC | NC | NC | NC | NC | NC |
| Loma Grande | 13.44 | −0.5 | 10.7 | NC | 17.99 | 14.63 | 309.22 | NC | NC | NC | NC | NC | NC | NC | NC |
| Los Campanos | NC | NC | NC | NC | 20.81 | 13.25 | 331.53 | NC | NC | NC | NC | NC | NC | NC | NC |
| Montebello | 9.99 | −0.85 | 6.74 | NC | 15.88 | 12.17 | 229.72 | NC | NC | NC | NC | NC | NC | NC | NC |
| Polo Nuevo | NC | NC | NC | NC | 21.55 | 14.06 | 493.67 | NC | NC | NC | NC | NC | NC | NC | NC |
| Ponedera | NC | NC | NC | NC | 16.61 | 10.05 | 469.95 | NC | NC | NC | NC | NC | NC | NC | NC |
| Puerto Giraldo | NC | NC | NC | NC | 19.32 | 10.94 | 337.1 | NC | NC | NC | NC | NC | NC | NC | NC |
| Repelón | 5.02 | −1.59 | 2.71 | NC | 13.95 | 9.3 | 448.94 | NC | NC | NC | NC | NC | NC | NC | NC |
| Sabanalarga | 8.94 | −0.87 | 6.27 | NC | 15.39 | 11.45 | 459.24 | NC | NC | NC | NC | NC | NC | NC | NC |
| San Jose | NC | NC | NC | NC | 21.16 | 12.85 | 237.3 | NC | NC | NC | NC | NC | NC | NC | NC |
| San Pedrito Alerta | NC | NC | NC | NC | 12.34 | 9.17 | 301 | NC | NC | NC | NC | NC | NC | NC | NC |
| Usiacurí | NC | NC | NC | NC | 17.2 | 11.84 | 462.17 | NC | NC | NC | NC | NC | NC | NC | NC |

Note(s): The NC cells indicate that it was not possible to obtain an estimate.

**Table A4.** Summary of parameters and AIC values of the different distribution functions using the method of moments fit.

| Rainfall Station | Gev | | | | Gumbel | | | Pearson Type III | | | |
|---|---|---|---|---|---|---|---|---|---|---|---|
| | $\alpha$ | k | u | AIC | $\alpha$ | u | AIC | $\alpha$ | $\lambda$ | m | AIC |
| Aeropuerto Ernesto Cortissoz | 14.83 | −0.09 | 7.78 | 624.06 | 7.99 | 16.82 | 630.19 | 0.05 | 1.28 | −6.74 | 612.91 |
| Candelaria | 18.36 | −0.05 | 7.16 | 289.34 | 19.75 | 7.34 | 289.42 | 0.05 | 1.80 | −15.25 | 286.15 |
| Casa de Bombas | 20.31 | −0.04 | 11.14 | 305.21 | 21.34 | 11.28 | 304.46 | 0.05 | 2.14 | −16.43 | 302.53 |
| El Porvenir | 30.62 | −0.03 | 16.27 | 282.17 | 31.86 | 16.46 | 281.09 | 0.04 | 2.28 | −26.85 | 280.26 |
| Hacienda el Rabón | 22.29 | 0.15 | 13.13 | 306.99 | 19.03 | 12.13 | 303.55 | 0.18 | 20.24 | −86.73 | 307.46 |
| Hibaracho | 23.76 | 0.09 | 19.56 | 425.27 | 21.34 | 18.93 | 421.96 | 0.11 | 9.03 | −51.00 | 425.15 |
| Las Flores | 13.55 | −0.03 | 5.01 | 252.52 | 14.05 | 5.09 | 251.47 | 0.09 | 2.34 | −14.39 | 250.78 |
| Lena | 20.89 | −0.08 | 11.30 | 435.40 | 23.37 | 11.58 | 437.99 | 0.04 | 1.40 | −10.43 | 428.84 |
| Loma Grande | 19.46 | 0.00 | 14.56 | 311.78 | 19.38 | 14.55 | 309.71 | 0.07 | 3.18 | −18.61 | 310.19 |
| Los Campanos | 24.30 | −0.03 | 12.40 | 334.71 | 25.15 | 12.54 | 333.74 | 0.05 | 2.37 | −22.57 | 332.62 |
| Montebello | 17.74 | 0.11 | 13.13 | 232.55 | 15.76 | 12.60 | 229.68 | 0.16 | 10.54 | −43.93 | 232.56 |
| Polo Nuevo | 23.09 | −0.08 | 12.78 | 494.24 | 25.89 | 13.09 | 496.74 | 0.04 | 1.38 | −10.94 | 485.21 |
| Ponedera | 18.25 | −0.15 | 7.60 | 467.98 | 23.55 | 7.81 | 479.47 | 0.03 | 0.59 | −1.78 | 434.16 |
| Puerto Giraldo | 22.97 | −0.06 | 0.56 | 339.93 | 24.87 | 9.81 | 340.72 | 0.04 | 1.73 | −17.78 | 336.73 |
| Repelón | 16.07 | 0.00 | 9.04 | 452.71 | 16.14 | 9.05 | 450.86 | 0.08 | 2.99 | −17.43 | 450.22 |
| Sabanalarga | 17.29 | −0.06 | 10.50 | 461.16 | 18.82 | 10.69 | 462.82 | 0.05 | 1.67 | −9.66 | 455.53 |
| San Jose | 22.4 | −0.07 | 11.87 | 235.42 | 24.63 | 12.14 | 238.90 | 0.04 | 1.56 | −13.14 | 235.42 |
| San Pedrito Alerta | 13.80 | 0.02 | 9.21 | 304.18 | 13.47 | 9.14 | 301.61 | 0.11 | 3.78 | −16.67 | 303.00 |
| Usiacurí | 10.09 | −0.07 | 10.66 | 463.42 | 21.25 | 10.91 | 465.99 | 0.04 | 1.44 | −9.56 | 456.32 |

**Table A5.** Distribution of best fit.

| Rainfall Station | Code | Best fit | AIC | Cumulative Precipitation during the 5 Days Prior to the Occurrence of the Annual Maximum Daily Precipitation (mm) | | | | | |
|---|---|---|---|---|---|---|---|---|---|
| | | | | RP (Years) | | | | | |
| | | | | 2.33 | 5 | 10 | 20 | 50 | 100 |
| Aeropuerto Ernesto Cortissoz | 29045020 | Pearson Type III-MM | 612.913 | 15.30 | 31.70 | 46.2 | 60.40 | 78.90 | 92.80 |
| Candelaria | 29040260 | Pearson Type III-MM | 286.15 | 17.10 | 36.30 | 52.50 | 68.20 | 88.20 | 103.0 |
| Casa de Bombas | 29030410 | Pearson Type III-MM | 302.531 | 22.40 | 43.00 | 60.20 | 76.50 | 97.30 | 113.0 |
| El Porvenir | 14010090 | Gumbel-ML | 279.434 | 32.70 | 57.30 | 77.30 | 96.50 | 121.00 | 140.0 |
| Hacienda el Rabón | 29040270 | Gumbel-MM | 303.552 | 23.10 | 40.70 | 55.00 | 68.70 | 86.40 | 99.70 |
| Hibaracho | 14010020 | Gumbel-MM | 421.96 | 31.30 | 50.90 | 66.90 | 82.30 | 102.00 | 117.0 |
| Las Flores | 29045120 | Gumbel-ML | 249.016 | 12.00 | 22.20 | 30.60 | 38.60 | 48.90 | 56.70 |
| Lena | 29040200 | Pearson Type III-MM | 428.843 | 22.10 | 44.90 | 64.80 | 84.10 | 109.00 | 128.0 |
| Loma Grande | 29030270 | Gumbel-ML | 309.216 | 25.00 | 41.60 | 55.10 | 68.10 | 84.80 | 97.40 |
| Los Campanos | 29040290 | Gumbel-ML | 331.534 | 25.30 | 44.50 | 60.10 | 75.10 | 94.50 | 109.0 |
| Montebello | 29040020 | Gumbel-MM | 229.684 | 21.70 | 36.20 | 48.10 | 59.40 | 74.10 | 85.10 |
| Polo Nuevo | 29040080 | Pearson Type III-MM | 485.207 | 24.70 | 49.90 | 72.00 | 93.60 | 122.00 | 143.0 |
| Ponedera | 29040070 | Pearson Type III-MM | 434.159 | 14.50 | 36.40 | 58.80 | 82.20 | 114.00 | 139.0 |
| Puerto Giraldo | 29040300 | Pearson Type III-MM | 336.73 | 21.90 | 46.10 | 66.70 | 86.50 | 112.00 | 131.0 |
| Repelón | 29037060 | Gumbel-ML | 448.938 | 17.40 | 30.20 | 40.70 | 50.70 | 63.70 | 73.50 |
| Sabanalarga | 29040190 | Pearson Type III-MM | 455.533 | 19.70 | 38.10 | 53.70 | 68.80 | 88.20 | 103.0 |
| San Jose | 29030140 | Pearson Type III-MM | 235.415 | 24.00 | 48.00 | 68.60 | 88.60 | 114.00 | 134.0 |
| San Pedrito Alerta | 29040310 | Gumbel-ML | 301.002 | 16.50 | 27.90 | 37.20 | 46.10 | 57.70 | 66.30 |
| Usiacurí | 29040240 | Pearson Type III-MM | 456.317 | 20.90 | 41.60 | 59.60 | 77.10 | 99.80 | 117.0 |

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
