# Peer review of "Probabilistic Approach to Determine the Spatial Distribution of the Antecedent Moisture Conditions for Different Return Periods in the Atlántico Region, Colombia"

_water, doi:10.3390/w14081217_

Round 1
Reviewer 1 Report
The paper is well structured and clear in its objectives and results. I suggest extending the bibliography by also adding the following papers which, although they refer to the calculation of the antecedent soil moisture to evaluate the trigger thresholds of landslides, are to be considered for the method they propose, to be recalled in a more general paragraph on the topic:
Manfreda S, Fiorentino M, Iacobellis V. 2005 - DREAM A distributed model for runoff, evapotranspiration, and antecedent soil moisture simulation. Advances in Geosciences, 2, 31-39. DOI: 10.5194 / adgeo-2-31-2005
Lazzari M., Piccarreta M., Ray L.R. Manfreda S. 2020 - Modeling Antecedent Soil Moisture to Constrain Rainfall Thresholds for Shallow Landslides Occurrence. In: RAM L.R., LAZZARI M., 2020. In: Landslides: Investigation and Monitoring. Ed. IntechOpen, London (UK). https://www.intechopen.com/chapters/72592
Lazzari M, Piccarreta M, Manfreda S. 2019 - The role of antecedent soil moisture conditions on rainfall-triggered shallow landslides. Natural Hazards Earth System Science, 1-11. https://doi.org/10.5194/nhess-2018-371
The geographic location in figure 1 must certainly be improved in quality, because it is now almost illegible.
Author Response
Please see enclosd file.

Reviewer 2 Report
TITLE:
Inapt. "Department" does not appear fitting the title. Remove it.
ABSTRACT:
Line 17: Remove "Department". And also throughout the MS. People know it is synonmous to district.
Line 22: Surprised that IDW performed better. Or diod the authors miss something in krigging especially? WHy was IDW the best, a short explanation here maybe can improve the argument.
Last sentence: Is not backed up by the analysis. How can such data be useful is very vague.
INTRODUCTION:
I would like the authors to have extensive sections on interpolation algorithms and also the probability distributions and their relevance to meteorological data.
Table 1: Souce can be mentioned after the caption.
STUDY AREA:
Merge the text in just one paragraph.
Line 78: Livestock is not a landuse.
Line 82-85: Cite. Mention type of climate here also and cite that as well.
Figure 1: Very poor. Need to improve. The authors can take help of a GIS person to make a really good map shwocasing the meteorological stations also. The backgrond image can be a DEM or land use land cover map. Columbia could be a small inset. Otherwise it becomes meaningless.
METHODOLOGY/RESULTS:
Figure 2. Delete
Data collection: Not as bullets. Spread it into just two paragraphs.
Figure 3: Must be improved.
Section 3.4: The authors have looely provided qualitative details of intyerpolation algorithms. They need to discuss the physical basis (mathematical) of these algorithms. Which software was used for spatiual interpolation and how were the input data set into these geostatistical algorithms. Kriging has so many subtypes. Which was used in this analysis and why?
RMSE: Why only RMSE? WHy not use realtive bias, correlation coefficient?
Figure 4: Caption can be "Comparison of spatial interpolation methods. Spatial distribution of precipitation by using (a) Spline; (b) Kriging; and (c) IDW"
Figure 4: I have resvervations in Fig 4b unless authors describe in detail the procedure. It is missing out too many things in here. Would be better to show all values as background raster for the entire study area with contours overlain.
Figure 5: Legend itself appears like a figure. Could be very small. Again I would request authors to see comments on Figure 4.
Table 6, 8: Could be supplementary.
Fig 7: Use lat-lon grid labels. labels are not legible. Remove outer neatline. Remove the word "legend". Remove "AMC". Have an interpolated raster in the background.
Fig. 8, 9, 10: See comment on Fig 7
I believe figures (7-11) can be merged as a single figure.
Fig 12. See comments to Fig 5.
DISCUSSION:
Poor. Need to completely rewrite/overhaul.
CONCLUSIONS:
Write as a paragraphs detailing the knowledge synthesis of this research. Reduce the text. Just one sentence on interpolation would do. Last bullet can be deleted.
Author Response
Please see enclosed file and the new version of the manuscript.
